# FAIRSKIN: FAIR DIFFUSION FOR SKIN DISEASE IMAGE GENEARTION

## ABSTRACT

Image generation is a prevailing technique for clinical data augmentation for advancing diagnostic accuracy and reducing healthcare disparities. Diffusion Model (DM) has become a leading method in generating synthetic medical images, but it suffers from a critical twofold bias: (1) The quality of images generated for Caucasian individuals is significantly higher, as measured by the Fréchet Inception Distance (FID). (2) The ability of the downstream-task learner to learn critical features from disease images varies across different skin tones. These biases pose significant risks, particularly in skin disease detection, where underrepresentation of certain skin tones can lead to misdiagnosis or neglect of specific conditions. To address these challenges, we propose FairSkin, a novel DM framework that mitigates these biases through a three-level resampling mechanism, ensuring fairer representation across racial and disease categories. Our approach significantly improves the diversity and quality of generated images, contributing to more equitable skin disease detection in clinical settings.

## 1 INTRODUCTION

Artificial intelligence (AI) is revolutionizing healthcare, particularly in medical imaging, where it enhances diagnostic accuracy and helps reduce healthcare disparities (Li et al., 2023; Liu et al., 2024; Bianchi et al., 2023; Akrout et al., 2023). One critical application of AI is image generation (Bianchi et al., 2023; Chen, 2023; Zhang et al., 2023; Hung et al., 2023; Lee et al., 2024), used for augmenting clinical data to improve disease detection, support rare condition diagnosis, and provide clinicians with more comprehensive insights. Among the leading models for synthetic medical image generation is the Diffusion Model (DM) (Akrout et al., 2023; Zhang et al., 2023), which has shown significant potential across various medical imaging tasks, including skin disease detection (Kazerouni et al., 2023; Benjdira et al., 2024; Groh et al., 2024).

However, despite these advancements, the use of Diffusion Models for medical image generation is hindered by significant bias. In this paper, we identify a twofold bias that limits the fairness and effectiveness of these models in clinical applications. **First**, the quality of images generated for African individuals is substantially lower, as evidenced by higher Fréchet Inception Distance (FID) scores, compared to images generated for other ethnicities. **Second**, downstream learners extract meaningful features from disease images of different skin tones with varying effectiveness, which in turn affects the accuracy of diagnosis.

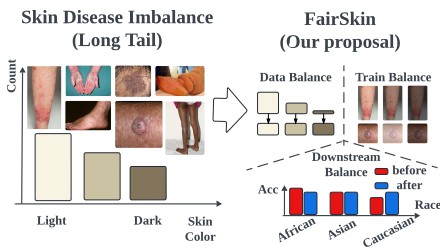

Figure 1: Overview of skin disease imbalance and the FairSkin framework: addressing long-tail distributions in skin disease data (Left) and improving fairness across racial groups through three-level resampling (Right).

These biases are particularly concerning in the detection of skin disease, where the appearance of conditions can vary significantly across skin tones. As demonstrated in Figure 1, certain skin tones are usually underrepresented and in the long tail of skin disease data, increasing the risk of misdiagnosis or delayed diagnosis, exacerbating existing health disparities (Khatun et al., 2024). Addressing these biases is crucial for ensuring that AI-driven diagnostic tools benefit all patient populations equitably.

To tackle these challenges, we propose `FairSkin`, a novel framework designed to mitigate racial biases in medical image generation using a three-level resampling mechanism. Our method employs ❶ balanced sampling and ❷ a class diversity loss during DM training, ensuring that underrepresented ethnic groups are fairly represented both in quantity and quality. In addition, we enhance downstream performance by ❸ employing imbalance-aware augmentation and dynamic reweighting techniques, promoting fairness in classification tasks.

The contributions of this work are as follows:

- **Bias identification**: We identify a critical twofold bias in Diffusion Models, favoring Caucasian individuals in both representation and image quality for skin disease detection. We analyze the underlying causes of this bias, highlighting data imbalances and the challenges in distinguishing between certain skin tones.
- **Fairness enforcement**: We introduce `FairSkin`, a novel framework that improves the diversity and quality of generated images for underrepresented groups, and ensures fairer classification performance across ethnicities.
- **Experiment validation**: We demonstrate through experiments that our framework significantly improves the fairness, quality, and diagnostic utility of generated medical images, promoting equitable healthcare outcomes.

## 2 RELATED WORK

### 2.1 IMAGE GENERATION FOR MEDICAL DISEASES

Image generation for medical diseases is a burgeoning field that leverages generative models, such as Generative Adversarial Networks (GANs) (Ma et al., 2021), Variational Autoencoders (VAEs) (Volokitin et al., 2020), and Diffusion Models (DMs) (Akrout et al., 2023), to create synthetic medical images. This technology facilitates addressing critical challenges in the medical domain (Kazerouni et al., 2023), such as the scarcity of labeled data, privacy concerns, and the lack of experts. Compared to GAN-based and VAE-based methods (Frid-Adar et al., 2018; Rais et al., 2024; Cetin et al., 2023), DM-based strategies have been widely utilized for data augmentation in the medical domain. For instance, Chen (2023) utilized DM for data augmentation for image classification of the Cell Cycle Phase, indicating a high potential in addressing issues related to insufficient data or unbalanced sample sizes. Recently, conditional diffusion models (Zhang et al., 2023) have gained significant attention in medical image generation for their flexibility and performance, achieving state-of-the-art results in tasks such as MRI (Dorjsembe et al., 2024), X-ray (Hung et al., 2023), and skin disease generation (Akrout et al., 2023; Borghesi & Calegari, 2024), and demonstrating comparable classification performance even with fully synthetic data. All the works have proved the effectiveness of DM in medical image generation. However, all the above approaches only consider generation quality, ignoring the bias existing in the generation process, which may limit the fairness for specific tasks.

### 2.2 BIAS IN IMAGE GENERATION

With the growing prevalence of generative artificial intelligence, concerns over bias in image generation have garnered significant attention due to its potential to influence social perceptions and cognition (Yang et al., 2024; Bragazzi et al., 2023). Recent studies have identified systematic gender, racial, and cultural biases in prominent image generation models such as GPT-4 (Waisberg et al., 2023), Stable Diffusion (Rombach et al., 2022b), and DALL·E 2 (Ramesh et al., 2021; Zhou et al., 2024). For instance, Zhou et al. (2024) reported gender disparities in occupational portrayals, with fewer female representations compared to males, and a racial bias favoring White individuals over Black individuals. Similarly, Wang et al. (2023) observed gender associations with personal interests, linking "science" with men and "art" with women. Moreover, cultural biases have been found, with images predominantly reflecting over-represented cultures like the United States over others (Basu et al., 2023). Such biases will lead to consequences, especially for skin disease detection, where underrepresentation of certain skin tones can lead to severe misdiagnosis (Babool et al., 2022; Pundhir et al., 2024). Recent efforts have focused on two main strategies: (1) weight refinement through fine-tuning or model editing (Jin et al., 2024), and (2) data reorgani-

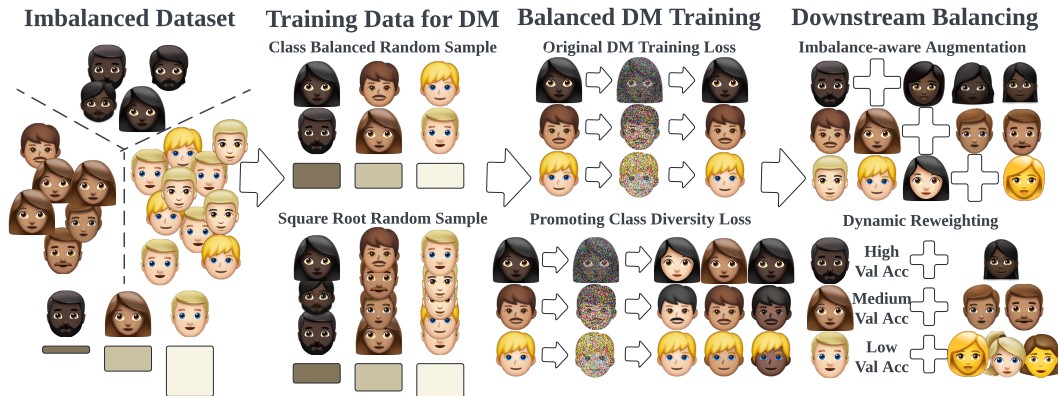

Figure 2: An overview of the `FairSkin` framework, illustrating the pipeline from an imbalanced dataset to balanced diffusion model (DM) training and downstream balancing. The process includes class balanced and square root random sampling methods for training data, balanced DM training incorporating class diversity loss, and downstream balancing through imbalance-aware augmentation and dynamic reweighting based on validation accuracy.

zation via prompt-based techniques (Wan & Chang, 2024), conditional generation (He et al., 2024), and re-sampling (Zameshina et al., 2023). Distinct from these methods, our `FairSkin` framework employs a comprehensive re-sampling strategy tailored to the skin disease, integrating specific considerations related to training data, objectives, and downstream tasks to achieve better fairness.

## 3 METHODOLOGY

### 3.1 PROBLEM SETUP

In this study, we address the generation and classification of skin diseases across different racial groups, specifically focusing on three racial categories: Asian, African, and Caucasian. The classification task involves five distinct skin diseases: Allergic Contact Dermatitis, Basal Cell Carcinoma, Lichen Planus, Psoriasis, and Squamous Cell Carcinoma. The dataset utilized in this study is denoted by $\mathcal{S}$, comprising $N$ samples. Each sample is represented as a triplet $(x_i, r_i, d_i)$, where

$$\mathcal{S} = \{(x_i, r_i, d_i)\}_{i=1}^{N},$$

with $x_i \in \mathcal{X}$ representing the input data (e.g., skin images), $r_i \in \mathcal{R}$ denoting the race class, and $d_i \in \mathcal{D}$ indicating the skin disease class of the $i$-th sample.

A significant challenge in this classification task is the imbalance in the distribution of samples across different race and disease classes. Let $N_{r,d}$ denote the number of samples belonging to race $r$ and disease $d$, where $r \in \mathcal{R}$ and $d \in \mathcal{D}$. The dataset exhibits imbalance in the following ways: **Firstly**, there is a *race-class imbalance*, meaning that the number of samples across different races is not uniform. For each race $r \in \mathcal{R}$, the total number of samples $N_r$ is given by $N_r = \sum_{d \in \mathcal{D}} N_{r,d}$. The distribution $\{N_r\}$ is imbalanced, i.e., there exist races $r_1$ and $r_2$ such that $N_{r_1} \neq N_{r_2}$. **Secondly**, within each race $r$, there is a *disease-class imbalance*, where the distribution of skin disease classes varies significantly. For each race $r$, the number of samples for disease $d$ is

$$N_{r,d} = |\{i \mid r_i = r, d_i = d\}|.$$

This variation leads to the underrepresentation of certain diseases within specific races, exacerbating the overall class imbalance in the dataset. The primary objective of this study is to develop a generative model that generates the skin disease image conditioned on both the race and disease classes. To achieve this, we propose the `FairSkin` framework, which integrates balanced sampling strategies and class diversity loss within a diffusion model (DM) to mitigate the identified imbalances. The framework also incorporates downstream balancing techniques to ensure fair classification performance across all races.

## 3.2 Training Data Resampling

To address the imbalance in the dataset, we employ two resampling strategies: Class Balanced Random Sampling (CBRS) and Square Root Random Sampling (SQRS). These methods aim to ensure that the training data fed into the diffusion model maintains a balanced representation across both race and disease classes.

**Class Balanced Random Sampling.** CBRS ensures that each class within the dataset is equally represented during the training process. To mitigate class imbalance, we introduce a reweighting technique where each sample is assigned a weight inversely proportional to its class frequency. Formally, for each $(r, d)$ pair, the weight $w_{r,d}$ is defined as: $w_{r,d} = \frac{1}{N_{r,d}}$, where $N_{r,d}$ is the number of samples belonging to race $r$ and disease $d$. During training, these weights are incorporated into the loss function to ensure that minority classes contribute proportionally more to the overall loss, thereby encouraging the model to learn equally from all classes.

Importantly, we utilize all available samples without restricting the number of samples per class based on the minimum class size. This approach allows the model to leverage the full diversity of the dataset while addressing imbalance through reweighting.

**Square Root Random Sampling.** SQRS is designed to balance the representation of classes by assigning weights based on the square root of their class frequencies. Specifically, for each $(r, d)$ pair, the weight $w_{r,d}$ is set to: $w_{r,d} = \frac{1}{\sqrt{N_{r,d}}}$, where $N_{r,d}$ is the number of samples belonging to race $r$ and disease $d$. This weighting scheme reduces the dominance of majority classes more gently compared to CBRS, allowing for a more nuanced balance between oversampling minority classes and retaining information from majority classes.

## 3.3 Balanced DM Training

Training DM in a balanced manner is pivotal to prevent the propagation of existing data imbalances into the generative process. This subsection outlines the original diffusion model training and promoting class diversity loss.

**Original Diffusion Model Training.** The original diffusion model training adheres to the standard methodology, wherein the model learns to reverse a predefined noise diffusion process to generate high-fidelity images. The training objective is to minimize the discrepancy between the generated images and the real images in the dataset. Mathematically, the loss function commonly used is the Variational Lower Bound (VLB), defined as:

$$\mathcal{L}_{\text{DM}} = \mathbb{E}_{x,\epsilon,t} \left[ \|\epsilon - \epsilon_\theta(x_t, t)\|^2 \right],$$

where $x$ represents the original data sample, $\epsilon$ is Gaussian noise added to $x$, $t$ denotes the timestep in the diffusion process, $x_t$ is the noised version of $x$ at timestep $t$, and $\epsilon_\theta$ is the neural network parameterized by $\theta$ that predicts the noise. The objective is to train $\epsilon_\theta$ such that it accurately predicts the noise $\epsilon$ added to $x$, thereby enabling the model to reconstruct the original image from the noised version during the reverse diffusion process.

**Promoting Class Diversity Loss.** To counteract potential mode collapse and ensure that the diffusion model generates a diverse set of images across all race and disease classes, we introduce a Class-Balancing Diffusion Model (CBDM) (Qin et al., 2023). CBDM incorporates a class diversity loss component $L_r$ designed to enforce equitable representation of each $(r, d)$ class during the image generation process.

The regularization term $L_r$ is formulated to promote class diversity by penalizing discrepancies in noise predictions across different classes. Specifically, $L_r$ penalizes the model if the noise estimated under the true class label $y$ significantly differs from the noise estimated under a randomly sampled class label $y'$. This mechanism discourages the model from becoming biased towards majority classes and encourages it to maintain consistency across all classes. The regularization term $L_r$ is defined as:

$$L_r(x_t, y, t) = \frac{t}{|Y|} \sum_{y' \in Y} \|\epsilon_\theta(x_t, y) - \epsilon_\theta(x_t, y')\|^2,$$

where $\epsilon_\theta(x_t, y)$ is the estimated noise given the noisy image $x_t$ and class label $y$, $\epsilon_\theta(x_t, y')$ is the estimated noise given the noisy image $x_t$ but treated as class $y'$, $t$ is a scaling factor proportional to the diffusion step, $|Y|$ is the number of classes.

By promoting the class diversity loss through CBDM, the `FairSkin` framework effectively mitigates class imbalance in the generative process, ensuring that the diffusion model produces a balanced and diverse set of images across all race and disease classes.

## 3.4 DOWNSTREAM BALANCING

After training the diffusion model, it is essential to ensure that the classification performance with generated data augmentation remains balanced across all race and disease classes. The `FairSkin` framework incorporates two downstream balancing techniques: Imbalance-aware Augmentation and Dynamic Reweighting. These strategies enhance the classifier's ability to perform equitably across different race and disease classes.

**Imbalance-aware Augmentation.** Imbalance-aware Augmentation leverages the trained diffusion model to generate synthetic samples for underrepresented $(r, d)$ classes, thereby augmenting the training dataset to achieve a more balanced distribution. This process involves generating a specified number of synthetic images conditioned on each minority class to compensate for their scarcity in the original dataset. Formally, for each minority $(r, d)$ pair, the diffusion model generates $m_{r,d}$ synthetic images $\hat{x}_i$ as : $\hat{x}_i = \text{DM}(z_i; r, d), \forall i \in \{1, \ldots, m_{r,d}\}$, where $z_i$ represents the random noise input to the diffusion model. The generated synthetic samples are then combined with the original dataset to form an augmented dataset $\mathcal{S}_{\text{aug}}$:

$$\mathcal{S}_{\text{aug}} = \mathcal{S} \cup \{(\hat{x}_i, r, d)\}_{i=1}^M,$$

where $M = \sum_{r,d} m_{r,d}$. The number of synthetic samples $m_{r,d}$ for each $(r, d)$ pair is determined based on the desired class distribution, typically aiming to balance the number of samples across all classes or to achieve a predefined ratio that mitigates the original imbalance.

This augmentation strategy enriches the training data with diverse examples from minority classes, enhancing the classifier's ability to generalize and perform uniformly across all classes. By introducing synthetic variability, the model gains exposure to a broader range of features representative of each class, thereby improving its fairness in classification tasks.

**Dynamic Reweighting.** Dynamic Reweighting dynamically adjusts the class weights during the training of the classification model based on validation performance metrics, considering we have train, validation, and test split. This technique ensures that classes with poorer performance receive higher emphasis, promoting balanced learning across all classes. As training progresses, after each epoch, the disease validation accuracy $A_r$ for each racial class is computed. The weights are then updated based on the inverse of the validation accuracy: $w_r = \frac{1}{A_r}$, This update rule ensures that classes with lower validation accuracy $A_r$ receive higher weights, thereby increasing their influence on the loss function during subsequent training iterations. Such dynamic reweighting strategy ensures that the classifier focuses more on classes that are underperforming, promoting balanced performance across all race and disease classes.

## 4 EXPERIMENTS

### 4.1 DATASETS AND METRICS

In this work, we evaluate the proposed approach based on fitzpatrick17k datasets, which contain 16,577 clinical images with Fitzpatrick skin type labels. We selected the top 5 disease classes from the Fitzpatrick17k dataset (Groh et al., 2021) and further divided them into 15 sub-classes based on race labels with at least 12 images and a maximum of 412 images per class as shown in Table 4 in Appendix A. We split each subcategory into training, validation, and test sets in an 8:1:1 ratio. We evaluate `FairSkin` using two sets of metrics: generation quality and fairness.

**Fréchet Inception Distance (FID)**: We use FID (Heusel et al., 2017) to evaluate the similarity between generated images and real images in the feature space. By using a pre-trained Inception

network to extract features from both real and generated images, a lower FID score indicates better image generation quality.

**FID Variance**: In addition, we calculated the FID values for each of the 15 subcategories and computed the variance, resulting in the FID Variance Score, which is used to indicate the fairness of the generated image quality.

**Inception Score (IS score)**: IS is a commonly used metric for evaluating the quality of images generated by models such as GANs. It assesses the model's performance by measuring both the quality and diversity of the generated images.

**Demographic Parity (DP)**: DP is a widely used fairness metric in classification tasks that evaluates whether the proportion of positive outcomes is the same across different demographic groups. Formally, DP is expressed as $\text{DP} = \sum_{z \in Z} \left| p(\hat{Y} = 1) - p(\hat{Y} = 1 \mid Z = z) \right|$, where $p(\hat{Y} = 1)$ represents the overall probability of a positive classification outcome, and $p(\hat{Y} = 1 \mid Z = z)$ denotes the probability of a positive classification outcome for demographic group $Z = z$. The summation of the absolute differences across all demographic groups $Z$ reflects the level of demographic parity, with smaller values indicating greater fairness. This approach to ensuring fairness aligns with the framework described in (Agarwal et al., 2018), which proposes a reduction method to address fairness in classification tasks by reducing the problem to a sequence of cost-sensitive classification problems, allowing for efficient and fair classification across various groups.

**Equity-Scaled Segmentation Performance (ESSP)**: ESSP is introduced in (Tian et al., 2024) as a fairness-aware metric for segmentation tasks. It adjusts traditional segmentation metrics by incorporating the disparity in performance across demographic groups. It is calculated as:

$$\text{ESSP} = \frac{\mathcal{A}(\hat{y}, y)}{1 + \Delta}$$

where $\mathcal{A}(\hat{y}, y)$ represents the segmentation accuracy, and $\Delta$ denotes the disparity in performance across different demographic groups. A lower $\Delta$ indicates more equitable performance across groups, resulting in a higher ESSP score. This metric allows for the evaluation of both overall performance and fairness in medical image segmentation. The segmentation accuracy $\mathcal{A}(\hat{y}, y)$ in this context is replaced with our classification validation accuracy, where $\hat{y}$ is the predicted label by the model and $y$ is the true label. $\Delta$ is actually the Demographic Parity among three different demographic groups (Asian, African, Caucasian) to better align with the fairness evaluation requirements of our classification tasks.

### 4.2 Implementation Details

In the generation setting, we primarily use Stable Diffusion v1-4 (Rombach et al., 2022a) as the backbone. Each method undergoes full fine-tuning on this model with a batch size of 8, a learning rate of 3e-7, and a maximum of 12,000 training steps. The dataset labels and the prompts used for generation are identical, such as "Asian people basal cell carcinoma." For each method, we generated 1,000 images per class across 15 categories and conducted subsequent evaluations on the generated data. The seed for each image was fixed to ensure reproducibility.

For the downstream task classifier setting, we use ViT-Base-Patch16-224 (Wu et al., 2020) as the backbone. We fully fine-tuned the classifier on the generated images from each method, using a learning rate of 1e-4 and training for 10 epochs, as we found that 10 epochs are sufficient and result in the best model fit. All training and testing were conducted on 8 A6000 Ada GPUs.

### 4.3 Main results

As shown in Table 1, we evaluate the proposed `FairSkin` compared with the previous SOTA method Class-Balancing Diffusion Model (CBDM) (Qin et al., 2023). Besides, we also compare Class Balanced Random Sampling (CBRS) and Square Root Random Sampling (SQRS) as additional baselines. Evaluation results are summarized in Table 1, where all models are compared under the same data augmentation numbers. The following observations can be drawn: ❶ Our model significantly outperforms other methods in terms of fairness in generated image quality. Specifically, the FID variance value decreased by **276.63** compared to the vanilla method, and by **18.63**

compared to CBDM. We achieve this by employing Training Data Resampling to provide a more equitable data distribution for African individuals, and by using Balanced DM Training to bring the quality of African-related images closer to those of Caucasian and Asian groups in high-dimensional distributions. For more detailed information, please refer to Table 2. ❷ In terms of fairness in the downstream classifier, our model also achieves SOTA performance. By using an additional dataset with more fairly generated images for data augmentation, the downstream model can more accurately focus on disease characteristics. Additionally, by automatically adjusting the racial composition of the augmented dataset based on the validation results for each racial group, we can selectively increase the sampling rate for underperforming groups, thus improving the model's performance on those groups and enhancing overall fairness.

Table 1: Comparison of our `FairSkin` method against baselines in both image generation tasks and downstream tasks, evaluated using the image generation metric FID, FID Variance, IS score, and the downstream classifier metrics DP, ESSP.

| Methods | ↓FID | ↓FID Variance | ↓DP | ↑ESSP | ↑IS |
|---|---|---|---|---|---|
| No Data Augmentation | - | - | 18.22 | 5.11 | - |
| Vanilla | 53.19 | 603.74 | 25.04 | 3.76 | 2.39 ± 0.47 |
| CBRS | **49.52** | 471.38 | 15.28 | 5.85 | 2.32 ± 0.47 |
| SQRS | 50.29 | 441.62 | 18.03 | 5.08 | 2.35 ± 0.47 |
| CBDM | 52.31 | 345.74 | 13.13 | 6.11 | 2.52 ± 0.37 |
| **FairSkin** | 52.28 | **327.11** | **9.95** | **7.78** | **2.52 ± 0.38** |

## 4.4 ABLATION

We use `FairSkin-SS`/`FairSkin-SW` to represent models where Training Data Resampling uses SQRS, Balanced DM Training uses CBDM, and Downstream Balancing uses Imbalance-aware Augmentation Resample/Dynamic Reweighting, respectively. We use `FairSkin-CS`/`FairSkin-CW` to represent models where Training Data Resampling uses CBRS, Balanced DM Training uses CBDM, and Downstream Balancing uses Imbalance-aware Augmentation Resample/Dynamic Reweighting, respectively. Similarly, we use `FairSkin-S`/`FairSkin-C` to represent models where Training Data Resampling uses SQRS/CBRS, Balanced DM Training uses CBDM, and Downstream Balancing uses random sampling, respectively.

**Different Models Exhibit Different Fairness on downstream tasks.** We initially investigated the impact of various approaches in augmenting the same classifier with an equivalent volume of supplementary data during training. As illustrated in Figure 3a and Figure 4, compared to the respective baselines, the classification performance of the classifier trained with our approach exhibits a marginal improvement in accuracy (ACC) and demonstrates a substantial enhancement in fairness relative to the other methods.

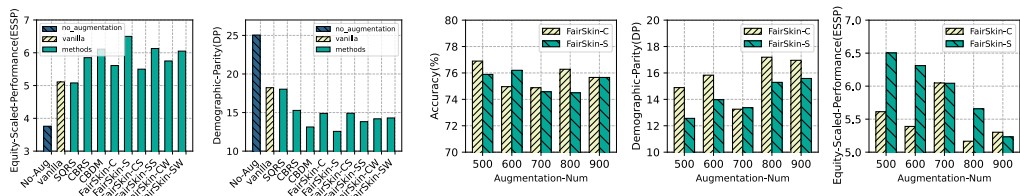

(a) Baselines vs `FairSkin`          (b) Performance across different augmentation sizes

Figure 3: **(a)** The comparison of `FairSkin` with baselines on downstream tasks. Under the condition of no data augmentation, the classifier exhibits the worst fairness. For other methods, we generate 7,500 images for data augmentation. `FairSkin` consistently demonstrates superior performance across various fairness metrics compared to other methods. **(b)** Variation in augmented dataset size. In this experiment, we provided an equal number of augmented images for each subcategory. Augmentation-Num refers to the number of augmented images per class. The results show that ACC, ESSP, and DP each have their own optimal number of augmented images.

**Different Sampling Number Leads to Different Fairness.** We evaluated the impact of adding varying amounts of additional dataset images across different methods. As shown in Figure 3b, we

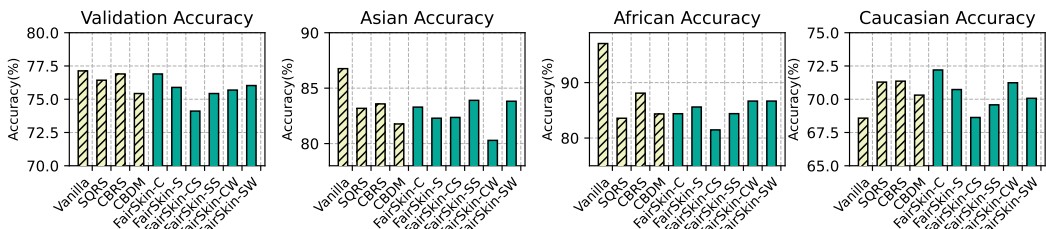

Figure 4: ACC scores under different methods. We evaluated the ACC for different racial groups as well as the overall ACC. Our method slightly reduced the ACC for groups with higher classification accuracy, but significantly improved the ACC for groups with lower classification accuracy, thereby enhancing fairness.

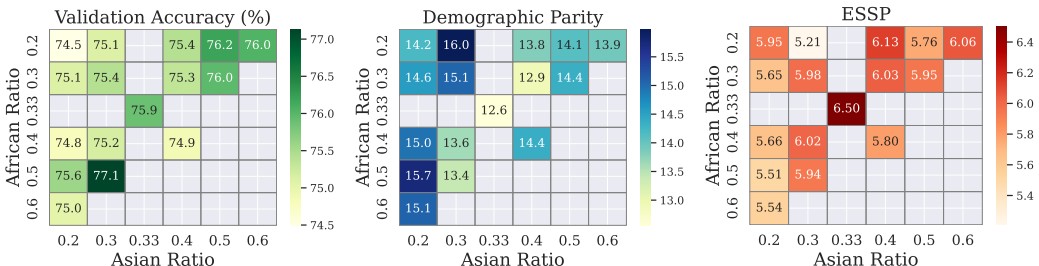

Figure 5: Ethnic proportion search for imbalance-aware augmentation when using FairSkin for downstream disease classification task. We fixed the total number of images for data augmentation at 7,500. All images were generated using the model trained with Training Data Resampling and Balanced DM Training. During the classifier training process, we maintained a fixed racial composition for data augmentation, for example, using a ratio of African:Asian:Caucasian = 0.3:0.2:0.5, which corresponds to 2,250:1,500:3,750 images, with an equal number of images for each disease type within each racial group. We calculated the classifier's ACC, DP, and ESSP.

searched for the optimal number of images within the range of 500 to 900 for each method and found that the ideal amount varies across different approaches. The results indicate that the number of images should neither be too large nor too small, as more sampling leads to convergent absolute downstream performance and exacerbated unfairness.

**Ethnic Proportion and Reweighting in Third-Level Resampling.** We attempted to modify the proportion of different ethnic groups in the additional dataset during the third-level resampling to further enhance the fairness performance of the classifier. As shown in Figure 5, we searched for the optimal proportion across different methods and observed that the performance of each method peaked at different proportions. In our approach, the proportion remains fixed throughout the classifier's entire training process. Furthermore, we compared the results of applying the reweighting method in the third-level resampling on top of different first- and second-level resampling strategies, as illustrated in Figure 6. The experimental results demonstrate that the reweighting method can improve fairness compared to using a fixed proportion at some data augmentation amounts.

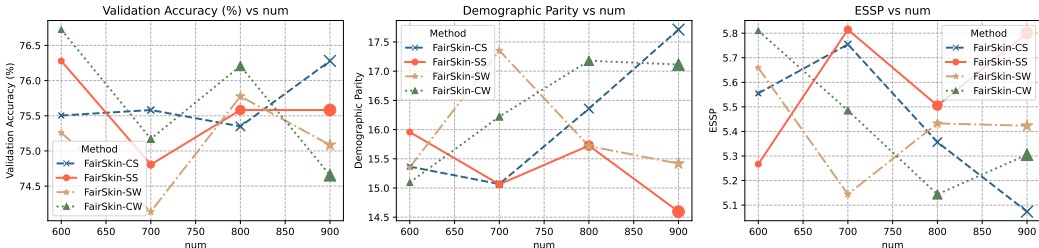

Figure 6: Dynamic reweighting effect for downstream disease classification tasks. In this experiment, we fixed the total number of data augmentation samples at 7,500 and applied Imbalance-aware Augmentation and Dynamic Reweighting separately to improve the performance of the classifier.

## 4.5 GENERATION PERFORMANCE

In this section, we first present the quality of the generated images. We evaluate the image quality by calculating the FID score. However, due to the small number of images per class in the original dataset, the FID score tend to be relatively high. Therefore, we focus only on the differences between the same class across different methods. Table 2 presents the FID results across different ethnic groups for various methods, as well as the overall FID score computed without subcategory distinction. Additionally, the variance of the FID scores across the three ethnic groups is reported to indicate the unfairness in the quality of the generated images for different ethnicities.

Table 2: Comparison of the performance of different methods on image generation quality. It is evident that while slightly enhancing the quality of Caucasian and Asian-related images, `FairSkin` significantly improves the quality of African-related images, leading to a substantial reduction in the FID variance.

| Methods | ↓Caucasian FID | ↓Asian FID | ↓African FID | ↓Variance | ↓Overall FID |
|---|---|---|---|---|---|
| Vanilla | 80.96 | 87.01 | 126.22 | 603.74 | 53.19 |
| CBRS | 81.60 | 90.51 | 122.86 | 471.38 | **49.52** |
| SQRS | 84.51 | **87.00** | 122.09 | 441.62 | 50.29 |
| CBDM | 80.86 | 88.95 | 116.34 | 345.74 | 52.31 |
| **FairSkin** | **79.67** | 88.63 | **114.50** | **327.11** | 52.28 |

By using a fine-tuned ViT-Base-Patch16-224 model, which achieves the best performance in disease classification in our settings, we extract image features and use them to compute the IS score. For each subcategory, we provide 1,000 images for IS score calculation and randomly select 3,000 images to generate t-SNE visualizations based on either race labels or disease labels, as shown in Table 3. It is evident that our method achieves higher IS scores compared to the baseline. Additionally, in the race label visualizations, we observe that the distribution of ethnic groups is more consistent, with no clear separations, suggesting our proposed `FairSkin` achieves a fairer generation in terms of different races. In the disease label visualizations, we find that the points corresponding to different diseases in our method are more densely clustered, indicating higher distinguishability.

Table 3: Visualization of t-SNE under the disease label and race label, respectively. The number refers to the corresponding IS score of generated images using different methods.

| Vanilla: 2.39 ± 0.47 | CBRS: 2.32 ± 0.47 | SQRS: 2.35 ± 0.47 | FairSkin-C: 2.51 ± 0.37 | FairSkin-S: 2.52 ± 0.38 |
|---|---|---|---|---|

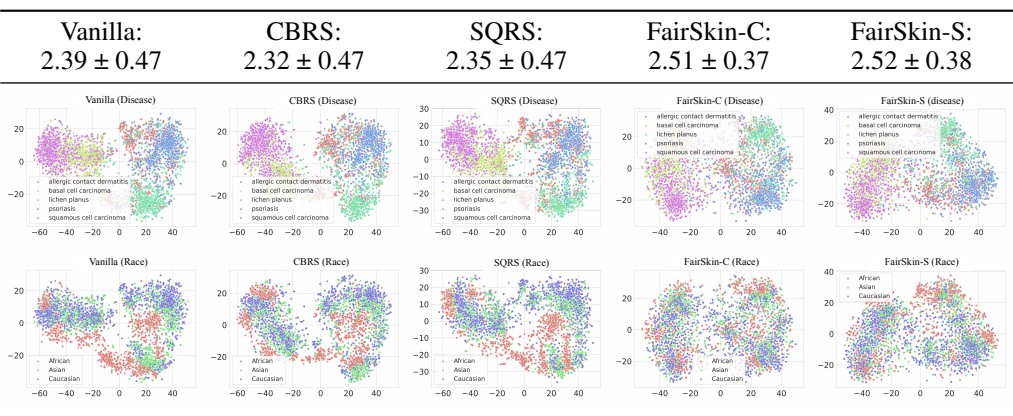

## 4.6 VISUALIZATION FOR DIFFERENT SD MODELS FOR MEDICAL FAIRNESS

As shown in Table 5 in Appendix B, we generated corresponding images for different disease types, ethnic groups, and methods, with the seed and prompt fixed for the images in each row. Even from the perspective of non-medical experts, it can be concluded that our method produces superior image generation results. These advantages include a more detailed depiction of body parts, closer alignment with human anatomical features, and more pronounced disease representations.

## 5 CONCLUSION

In this paper, we address the challenge of balancing fairness and performance in image generation and downstream classifier tasks when dealing with long-tailed datasets through `FairSkin`. We introduce a novel framework and propose strategies at three levels: Training Data Resampling, Balanced DM Training, and Downstream Balancing. Our approach effectively enhances the overall quality of generated images while reducing disparities in image generation quality across different classes. Furthermore, it improves the fairness of downstream classifiers when using generated images for data augmentation. Our comprehensive experiments demonstrate significant performance improvements across various tasks, highlighting the practicality and effectiveness of our methods in real-world applications.

## 6 BROADER IMPACT AND FUTURE WORK

The `FairSkin` framework has the potential to significantly impact the development of AI-driven medical diagnostic tools by addressing racial bias in medical image generation. By ensuring fair representation and improved image quality across diverse skin tones, this work can help reduce misdiagnoses and healthcare disparities, particularly in dermatology. The approach presented here contributes to broader efforts in making AI technologies more inclusive and equitable, promoting better healthcare outcomes for underrepresented groups.

In terms of future work, we aim to extend `FairSkin` to other domains of medical imaging beyond dermatology, such as radiology and ophthalmology, where racial and ethnic disparities in diagnostic performance have also been observed. Additionally, we plan to refine the resampling techniques and class diversity loss functions to further enhance fairness and representation. Expanding the framework to include real-world clinical validation will also be a key focus to ensure its effectiveness in practical healthcare settings.

## 7 REPRODUCIBILITY STATEMENT

To ensure the reproducibility of our results, we provide detailed descriptions of our experimental setup, model architecture, and training procedures in the subsection 4.2. This includes all hyperparameters, and data preprocessing steps. Additionally, the source code and scripts for reproducing our experiments will be made publicly available, along with the configurations necessary to replicate our findings. The datasets utilized in this study are publicly available and properly cited in the reference, ensuring that other researchers can easily access and validate our work.

## 8 ETHICS STATEMENT

Our research focuses on generating fair and diverse skin disease images to address disparities in medical diagnostics for underrepresented populations. We ensure that various skin tones, particularly those underrepresented in existing datasets, are adequately reflected, thereby mitigating diagnostic bias. Our synthetic images are created without using personal data, ensuring compliance with privacy standards such as the Health Insurance Portability and Accountability Act (HIPAA) of 1996 and the General Data Protection Regulation (GDPR) (Regulation (EU) 2016/679). Furthermore, any generated images do not correspond to real individuals and are purely synthetic, further mitigating any potential privacy concerns. The methodology, including model architecture and data handling, is made transparent and reproducible. No real human subjects are involved, eliminating any risk of harm. Additionally, we recognize the potential for misuse of this technology and advocate for its responsible and ethical application in healthcare, ensuring that it is only applied in ways that enhance patient care and public health. Our work is committed to promoting health equity, improving diagnostic fairness, and advancing inclusive healthcare solutions.

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

# APPENDIX

**Organization.** The appendix is organized as follows. Section A describes the Dataset Details. Section B presents additional experimental results.

## A  DATASET DETAILS

Table 4 shows the sample count per disease and per racial group. As we can see, regardless disease types, Caucasian people have more samples than Asian and African people have the fewest sample counts.

Table 4: From the Fitzpatrick17k dataset, we selected five disease types that are common across three racial groups, resulting in a total of 15 subcategories.

|  | Squamous Cell Carcinoma | lichen planus | psoriasis | allergic contact dermatitis | basal cell carcinoma |
|---|---|---|---|---|---|
| Caucasian | 329 | 181 | 412 | 295 | 302 |
| Asian | 166 | 183 | 145 | 108 | 154 |
| African | 56 | 120 | 87 | 25 | 12 |

## B  ADDITIONAL EXPERIMENTAL RESULTS

Table 5 visualizes generated skin disease images across different racial groups using stable diffusion models by different methods. As we can see, our methods (FairSkin-C and FairSkin-S) show more details of diseases compared to other baselines.

Table 5: Visualization of medical fairness across different Stable Diffusion models. We present the generated images from the perspectives of disease types, ethnic groups, and methods.

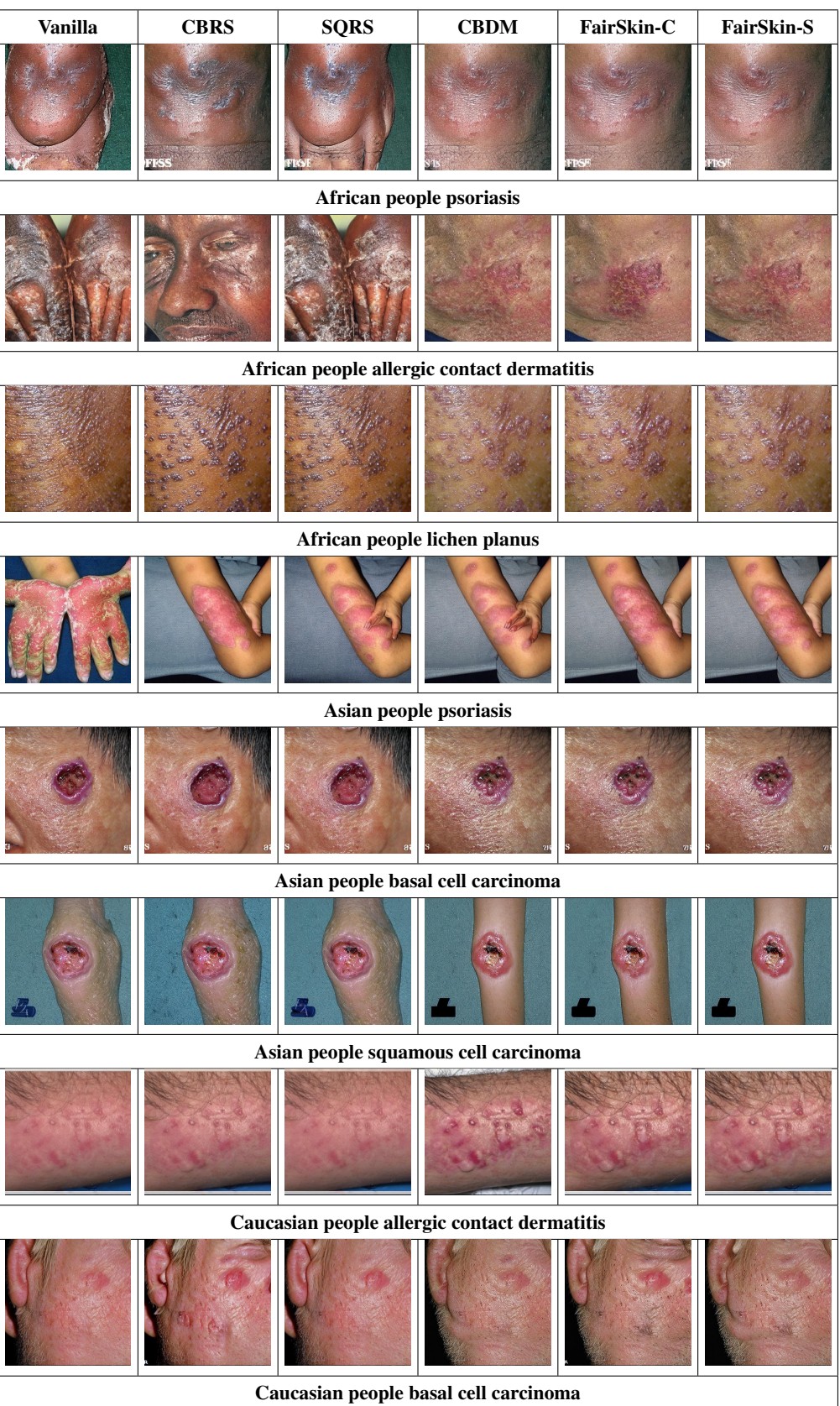

