# OpenReview forum: "FairSkin: Fair Diffusion for Skin Disease Image Geneartion"
_ICLR.cc/2025/Conference — ICLR 2025 Conference Withdrawn Submission_

### Official Review · Reviewer_WFDZ · 2024-10-18

**Soundness:** 2
**Presentation:** 3
**Contribution:** 2
**Rating:** 3
**Confidence:** 4

**Summary:**

The paper proposes a framework of a) diffusion-based data generation that is then used to b) augment the training data for c) skin disease classification. The authors use one dataset, the Fitzpatrick 17k and choose 5 disease classes and 3 skin type groups for their analyis. They report both performance and fairness metrics for the generation and the downstream classification task.

In their framework, the authors use four balancing strategies:
 * While training the generator: 1) Balanced data sampling and  2) using a class diversity loss.
 * While training the classifier: 3) Balanced data sampling and 4) reweighting the update step

**Strengths:**

This paper takes on a very important topic, which is bias in ML model performance due imbalances in the dataset. Ensuring good performance even at the long tail of both diseases and demographics is a difficult and worthwile task.
A list of merits:
1) The paper looks at bias mitigation methods in various stages: Data sampling for the augmentation data generator, Loss functions for the augmentation data generator, Data sampling for the downstream task, dynamic reweighting for the downstream task.
2) The paper presents ablation results on the effect of these various components.
3) A range of appropriate metrics are used for performance, and fairness for each subtask.
4) The used dataset is appropriate, contains difficult real world images and is a good example for the data availability bias in ML dermatology.
5) The paper is well organised and general also well written
6) There is a large range of analyses (ablations, tables, t-SNE, qualitative images)
7) The introductory parts explain the metrics, samplining and loss adaptations well.

**Weaknesses:**

This paper falls between a methodological paper and an application paper.
For a methods paper, the individual methods (balanced sampling methods and losses) are either reused from other papers, or not substantial enough.
If we consider the whole framework as a creative combination of those methods, there are still problems: There is hardly any comparison to work by other authors, and the whole analysis is conducted on a single dataset.
For an applications paper I would also expect the use of more than one dataset, more (fairness) metrics and a deeper discussion of the clinical relevance of the findings.
The very thing the paper tries to solve also very much impedes it's impact: The scarcity of darker skin types in the dataset leads to a small test set und thus results from which it is unclear whether they generalise well. I encourage the authors to think about using additional attributes to repeat and strenghten their analysis (e.g. using FairSkin on age/gender and seeing whether the results still hold).

A more detailed list of issues:
 1) The main claim of the paper is that it improves fairness by balancing out the lack of high quality training data of darker skin-types in the dataset. But the results in Fig 4 show that the accuracy increases for 'Caucasian' and decreases for 'Asian' and 'African' compared to the vanilla approach. So either I misunderstand something fundamentally, or the validation/test set is way too small to be representative, or this paper achieves the opposite of what it tries to do.
 2) Much of the analysis hinges on how the method does on the most underrepresented data group, which according to table 4, is 'African'. Given that the smallest disease subgroup only contains 12 examples and a 8:1:1 train/val/test split, some of the analysis is done on 1-2 datapoints. I would expect repetitions of results using different hold out sets to mitigate against this very low number.
 3) A repetition of (part of) the analysis on a different dataset would make this paper much stronger. There is e.g. the scin dataset, which also contains Fitzpatrick labels: https://github.com/google-research-datasets/scin
 4) A tabular ablation study would be really helpful to decide which of the components are actually making a difference. The current solution with FairSkin-XYC is hard to parse and read and I had to skip back and forth numerous times.
 5) Many of the claims in the conclusion are weekly supported by the experimental results:
 *  A):"Our approach effectively enhances the overall
quality of generated images while reducing disparities in image generation quality across different
classes." The overall quality as measures by IS and FID scores stays practiacally the same, and is worse than the baseline CBDM. But I agree with the second half of the statement.
 * B) "we address the challenge of balancing fairness and performance in image generation
and downstream classifier tasks" There is the compound ESSP score, but for example table 1 lacks an accuracy score altogether. There is also little discussion on how the balance the two objectives.
 * C) ..." tasks when dealing with long-tailed datasets [plural!] through FairSkin." But the paper only deals with one single dataset, and apparently only with a fixed datasplit within that dataset. The results would be much stronger if the framework was tested on different datasets too, even synthetic ones, or at least multiple ways to hold out data in the splits.
CBDM
* D) "Furthermore, it improves the fairness of downstream classifiers when using generated images for data augmentation" While the paper demonstrates an improvement in the Demographic Parity metric, it actually decreases downstream performance (fig 4) of the skin type classes with less images in the original training data. It makes me wonder whether DP is the appropriate fairness metric in this case. The paper lacks a discussion of why DP was chosen as the only fairness metric. Why did you e.g. not report equalized odds? Consider reporting more metrics and more results on a skin type and disease stratified level (in the appendix).
 * E) "Our comprehensive experiments demonstrate significant performance
improvements across various tasks" The validation accuracy stays practically the same. DP and FID variance improve, but again 8D) is this even meaningful when it seems to decrease performance of classes with least training data?
 * F) "Highlighting the practicality and effectiveness of our methods in
real-world applications." There is no discussion in the paper on how this could be practical or effective in real-world applications. Such a claim would at the very least require use test-data from a different dataset. Or an analysis on most-confounded disease classes to look at potential real-world harm of misclassifications.
6) I would have appreciated a deeper analysis stratified by disease to see how the method fares for diseases with more or less training data (for individual/all skin type classes)
7) There is very little discussion in the introduction and elsewhere about fairness in ML literature.


A general remark:
* The paper indiscriminately uses the words racial/ethnic/demographic classes when referring to the above labels. Race, ethnicity, and demographic, are a) not synonyms and b) not covered by the Fitzpatrick scale. The Fitzpatrick scale measures skin type (how much tanning occurs under UV exposure).

**Questions:**

1) The authors use the Fitzpatrick17k dataset. How did they arrive at the labels 'Caucasian', 'Asian', and 'African'? The Fitzpatrick scale in that dataset is labelled with numbers 1-6.
2) In 4.5. the paper states: "In the disease label visualizations, we find that the points corresponding to
different diseases in our method are more densely clustered, indicating higher distinguishability." I disagree with that statement and think that the disease clusters look more muddled. Am I looking at the wrong image?
3) The ablations with number of augmentations show a wilde variance with no clear trends. This is a) not very informative and b) makes me wonder whether thanks to the small test-set and single repetion of experiments the results are only anectodal. Could you clarify what you were trying to express with these ablations and how one should choose the number of augmentations in practice?
4) In table 1, it is unclear which sampling strategy is used for FairSkin, is it CBRS or SQRS?

---

### Official Review · Reviewer_KWQg · 2024-11-03

**Soundness:** 2
**Presentation:** 2
**Contribution:** 1
**Rating:** 3
**Confidence:** 4

**Summary:**

The work proposes a diffusion model based approach to synthesizing skin images in an effort to mitigate skin tone and disease class biases in skin image datasets. The diffusion model synthesizes images conditions on skin tone and disease label. The paper proposed several resampling and weighting procedures to reduce the bias. The paper also incorporates a class diversity promoting loss into the model training.

**Strengths:**

Tackles an important topic, fairness/bias in medical (specifically skin) image analysis applications.

The work examined different classification and fairness performance measures.

**Weaknesses:**

None of the ideas presented in the methods is particularly exciting or novel.

The work fails to cite many recent works that tackled the problem of fairness in dermatology images.

There is no comparison with other competing methods (e.g. [5,6]).

This work synthesizes medical images, which, even though they look realistic, are not grounded in a real underlying physical manifestation of skin lesion pathology. Therefore, there is a risk of harm in using these images for serious medical applications. Medical experts have not assessed the quality of the synthesized images. The authors should clearly mark the synthesized images as such in all the figures.

**Questions:**

The authors do not mention how the "racial groups": Caucasian, Asian, and African are created from the Fitzpatrick17k dataset, since the dataset only contains Fitzpatrick skin tone (FST) labels. It appears that the authors make a huge leap of assumption by mapping FST labels to "race" thus conflating skin tone with race, but this assumption is scientifically incorrect and should be flagged for an ethics review. FST was originally proposed to describe the propensity of the skin to burn from UV light exposure, and should not be conflated with race or ethnicity [1], with Fitzpatrick himself stating that "ethinicity or "race" is only a cultural and political term with no scientific basis" [2]. For example, a study of 556 participants from South Africa showed the presence of all 6 FST labels [3], invalidating that assumption that Africans can be limited to only some FSTs.

This paper fails to cite a number of papers from the existing literature that deal with similar topics: skin lesion image synthesis for fairer diagnosis [5, 6] and a larger body of literature on dealing with fairer diagnosis methods for dermatology [7, 8, 9, 10, 11, 12, 13].

The demographic parity (DP) metric used to assess fairness is limited to scenarios where there are only 2 demographic groups [4]. Metrics such as the normalized accuracy range (NAR) [5] have been proposed to deal with scenarios with multiple groups.

Using the DP metric also "requires equal proportion of positive predictions in each group" [4], which is not the case with this work as shown in Table 4.

There is no comparison with other competing methods (e.g. [5,6]).

Why is the schematic of Fig 1 (in the lower right bar chart) showing the accuracy of African diminished in the blue after? I think you need to swap African with Caucasian.  But even then, is this schematic making the point that improving fairness has to come at the expense of lower accuracy for some groups?

L056-058: How is the "balanced" in point (1) different from the "fairly represented" in point (2)?

L151: Having Nr1 \neq Nr2 as the mathematical definition for imbalance seems too structured, e.g. if there 1000 images of light skin and 1001 of dark skin, I don't agree to declaring the data as biased because 1000 \neq 1001.

L171: resampling inversely proportional to frequency is a standard approach.

L171 and L181: 1/N vs 1/sqrt(N) is a trivial point not worthy, in reviewer's opinion, of have two sections to cover them. You could have written: W = N^p, where p = -1 , -1/2 , etc.

L204: Mode collapse is only mentioned here without any explanation or introduction.

L206: The authors write "we introduce CBDM" but then write that this approach of using class diversity loss Lr is existing work. So what is novel here? Is it only the application to skin? This must be clarified in the writing.

L215: How can calculating the subtraction between two noise signals be a robust measure given the randomness of the noise;  why isn't a statistical measure of the noise used instead, e.g.,  Lr = abs ( variance ( noise for class y )   minus  variance ( noise for class y' ) ).

L270: justify why a pre-trained network on non skin images should be used to measure FID for skin images?

L276:  justify why "IS" is a suitable quality metric specifically for skin images?

L323:   "Vanilla" is not described.

The equations should be numbered.

[1] O. R. Ware et al., "Racial Limitations of Fitzpatrick Skin Type", Cutis 105.2 (2020): 77-80, https://www.mdedge.com/dermatology/article/216697/pigmentation-disorders/racial-limitations-fitzpatrick-skin-type.

[2] T. B. Fitzpatrick et al., "The Validity and Practicality of Sun-Reactive Skin Types I Through VI", Archives of Dermatology, 1988, https://doi.org/10.1001/archderm.1988.01670060015008.

[3] M. Wilkes et al., "Fitzpatrick Skin Type, Individual Typology Angle, and Melanin Index in an African Population", JAMA Dermatology, 2015, https://doi.org/10.1001/jamadermatol.2015.0351.

[4] A. Fraenkel, "Fairness and Algorithmic Decision Making: 5.3.1 Demographic Parity", https://afraenkel.github.io/fairness-book/content/05-parity-measures.html#demographic-parity.

[5] A. Pakzad et al., "CIRCLe: Color Invariant Representation Learning for Unbiased Classification of Skin Lesions", ECCV ISIC, 2022, https://doi.org/10.1007/978-3-031-25069-9_14.

[6] I. Ktena et al., “Generative models improve fairness of medical classifiers under distribution shifts”, Nature Medicine, 2024, https://doi.org/10.1038/s41591-024-02838-6.

[7] N. Bayasi et al., “BiasPruner: Debiased Continual Learning for Medical Image Classification”, MICCAI, 2024, https://doi.org/10.1007/978-3-031-72117-5_9.

[8] Ghadiri et al., “XTranPrune: eXplainability-Aware Transformer Pruning for Bias Mitigation in Dermatological Disease Classification”, MICCAI, 2024, https://doi.org/10.1007/978-3-031-72117-5_70.

[9] Y. Guo et al., “FairQuantize: Achieving Fairness Through Weight Quantization for Dermatological Disease Diagnosis”, MICCAI, 2024, https://doi.org/10.1007/978-3-031-72117-5_31.

[10] Aayushman et al., “Fair and Accurate Skin Disease Image Classification by Alignment with Clinical Labels”, MICCAI, 2024, https://doi.org/10.1007/978-3-031-72378-0_37.

[11] C. Chiu et al., “Achieve fairness without demographics for dermatological disease diagnosis”, Medical Image Analysis, 2024, https://doi.org/10.1016/j.media.2024.103188.

[12] C. Chiu et al., “Toward Fairness Through Fair Multi-Exit Framework for Dermatological Disease Diagnosis”, MICCAI, 2023, https://doi.org/10.1007/978-3-031-43898-1_10.

[13] S. Du et al., “FairDisCo: Fairer AI in Dermatology via Disentanglement Contrastive Learning”, ECCV ISIC, 2022, https://doi.org/10.1007/978-3-031-25069-9_13.

**Details Of Ethics Concerns:**

See last point under weaknesses.

---

### Official Review · Reviewer_JohS · 2024-11-03

**Soundness:** 2
**Presentation:** 3
**Contribution:** 2
**Rating:** 3
**Confidence:** 4

**Summary:**

This paper proposes FairSkin, a diffusion model (DM) framework designed to address biases in synthetic skin disease image generation for underrepresented groups. The work is motivated by observed disparities in image quality across skin tones, with minority skin tones often receiving lower-quality synthetic images and being more susceptible to misdiagnosis by downstream classifiers. FairSkin addresses these challenges with a three-tier approach involving balanced data resampling, a class diversity loss during DM training, and a downstream classifier balancing mechanism. Experiments on the Fitzpatrick17k dataset demonstrate that FairSkin outperforms baseline models on metrics for both image quality and fairness.

**Strengths:**

- The motivation is clear and relevant, as few works currently address fairness in diffusion model image quality specifically for skin lesion analysis.
- The writing is clear and well-organized, and the paper includes an extensive set of experiments that enhance the results and conclusions.

**Weaknesses:**

- Several existing studies apply diffusion models to fairness-related challenges in skin lesion imaging, and some even use the same Fitzpatrick17k dataset. The paper should compare the proposed generation method with these closely related works and clarify why it is a valuable addition to this research area.

    [1] Sagers, L.W., et al., Improving dermatology classifiers across populations using images generated by large diffusion models, NeurIPS Workshop, 2022.

    [2] Sagers, L.W., et al., Augmenting medical image classifiers with synthetic data from latent diffusion models, 2023.

    [3] Wang, J., et al., From Majority to Minority: A Diffusion-based Augmentation for Underrepresented Groups in Skin Lesion Analysis, MICCAI ISIC Workshop, 2024.

    [4] Ktena, I., et al., Generative models improve fairness of medical classifiers under distribution shifts, Nature Medicine, 2024.


- The proposed reweighting and resampling techniques are standard practices for handling imbalanced datasets and could be considered intuitive; thus, the novelty of the proposed method may be limited.


- Despite the class diversity loss, the generated images still lack diversity, which might limit their utility in augmenting datasets for varied skin lesion types.


- While demographic parity (DP) is a commonly used metric, it may not fully capture the nuances of fairness, as equal positive rates across groups do not necessarily indicate equitable outcomes. True positive rates across groups might provide a more accurate assessment of fairness in clinical applications. Both fairness metrics used in this study are DP-based, and the work should consider additional, informative metrics.

**Questions:**

- The study groups data by race; however, the Fitzpatrick17k dataset categorizes images by skin tone rather than racial background. There may be a misconception in equating skin tone with racial identity, which should be clarified in the paper.

- Each group has at most 412 images. Calculating FID on such small sets may yield unreliable results, as FID typically requires larger datasets for accurate assessments of image quality.

---

### Official Review · Reviewer_ixvc · 2024-11-03

**Soundness:** 2
**Presentation:** 3
**Contribution:** 2
**Rating:** 3
**Confidence:** 3

**Summary:**

The paper's primary objective is to balance fairness and model performance during the development of a generative model that generates the skin disease image conditioned on both race and disease classes. To mitigate the identified imbalances, they propose the FairSkin framework, which integrates balanced sampling strategies and class diversity loss within a diffusion model. They also consider two downstream balancing techniques, imbalance-aware augmentation, and dynamic reweighing, to address the performance-fairness trade-off.

**Strengths:**

- The authors propose a methodology for skin image generation based on diffusion models that account for fairness during image generation. To address the imbalance of skin lesions, they investigate two data sampling strategies, a balanced diffusion model training by introducing a class diversity loss, and downstream task balancing. All steps are clear and easy to follow.

- Several ablations are provided about the proposed pipelines and schemes. All figures and plots are clear and easy to understand.

- Their analysis considers three race groups and five diseases. The approach approach appears to enhance the overall quality of generated images while reducing disparities in image generation quality across different classes

**Weaknesses:**

- The paper's selection of three racial categories and five specific skin diseases lacks adequate justification. A more robust approach would involve a fine-grained control of skin tone, potentially incorporating established skin tone scales such as  the Fitzpatrick or Monk scales.
- The evaluation protocol's reliance on FID and Inception-based scores may not be appropriate in this context. While monitoring the generation performance of synthetic data is essential, utilizing these metrics as quality indicators is questionable, mainly because the Inception Score is based on ImageNet statistics, which may not be relevant to the datasets employed in this study.
- The paper raises concerns regarding dataset imbalance. However, it employs standard accuracy rather than a balanced accuracy metric, which could lead to skewed performance assessments, especially in imbalanced datasets. Balanced accuracy metric would provide a fairer evaluation of the model’s performance across underrepresented classes.
- The experimental protocol could also be enhanced by including additional datasets beyond Fitzpatrick17. Expanding the dataset selection would improve the generalizability of the findings. See some suggestions in the section below.

**Questions:**

### Suggestions/Questions

- A great addition to the work is to create a figure summarizing all contributions within the classification pipeline. The figure would help readers better navigate the paper and provide rapid reference across sections.
- In addition to Fitzpatrick17, several datasets with annotated skin tone information could enhance the study’s experimental scope. Suggestions Slice-3D [1], Casual Conversations [2], and FACET [3], each of which provides extensive metadata that could help to better condition image generation and studies about the impact of meta-data available with the image generation quality. With access to more datasets, a cross-dataset transfer evaluation could also be considered. This approach would involve training a diffusion model on one or more datasets and generating new data for a target dataset excluded from the training process, enabling exploration of the model’s generalization capabilities and potential for transfer learning.
- Investigate other fairness metrics, such as equal opportunity and disparate impact. Others, such as worst group statistics used in domain generalization, appear to be a great addition here.
- The results section lacks a comparative analysis with SOTA methods, particularly those designed to address imbalanced datasets. Table 1 should include comparisons with SOTA methods that deal with imbalance-related challenges. Works referenced in [1,2,3,4] may provide valuable benchmarks and ensure a more comprehensive evaluation.
- "Image Geneartion" -> "Image Generation" in the paper's title?

---

### Note · Authors · 2024-11-20

I have read and agree with the venue's withdrawal policy on behalf of myself and my co-authors.